# Two central pattern generators from the crab, *Cancer borealis,* respond robustly and differentially to extreme extracellular pH

Jessica A Haley[†], David Hampton, Eve Marder*

Volen Center and Biology Department, Brandeis University, Waltham, United States

**Abstract** The activity of neuronal circuits depends on the properties of the constituent neurons and their underlying synaptic and intrinsic currents. We describe the effects of extreme changes in extracellular pH – from pH 5.5 to 10.4 – on two central pattern generating networks, the stomatogastric and cardiac ganglia of the crab, *Cancer borealis.* Given that the physiological properties of ion channels are known to be sensitive to pH within the range tested, it is surprising that these rhythms generally remained robust from pH 6.1 to pH 8.8. The pH sensitivity of these rhythms was highly variable between animals and, unexpectedly, between ganglia. Animal-to-animal variability was likely a consequence of similar network performance arising from variable sets of underlying conductances. Together, these results illustrate the potential difficulty in generalizing the effects of environmental perturbation across circuits, even within the same animal.
DOI: https://doi.org/10.7554/eLife.41877.001

*For correspondence:
marder@brandeis.edu

Present address: [†]Neurosciences Graduate Program, University of California, San Diego, La Jolla, United States

## Introduction

Nervous systems must be both robust and adaptable to changes in internal and external conditions. Many intertidal marine crustaceans, such as the crabs and lobsters inhabiting the North Atlantic, experience large fluctuations in ocean temperature, acidity, dissolved oxygen levels, and salinity. The Jonah crab, *Cancer borealis,* can often be found foraging for food in intertidal zones where it experiences temperatures between 3°C and 24°C with fluctuations as great as 10°C in a single day (*Donahue et al., 2009*; *Haefner, 1977*; *Stehlik et al., 1991*). As pH is temperature-dependent, ocean pH fluctuations occur over the daily, monthly, and yearly experiences of long-lived crustaceans, such as *C. borealis.*

In several marine invertebrates including the lobster, *Homarus americanus,* and the crab, *Carcinus maenas,* hemolymph pH varies inversely with temperature following the rules of constant relative alkalinity (*Dove et al., 2005*; *Howell et al., 1973*; *Qadri et al., 2007*; *Truchot, 1978*; *Truchot, 1986*; *Wood and Cameron, 1985*). In other words, as temperature increases, hemolymph acidifies by approximately $-0.016$ pH/°C to maintain a constant ratio of pH to pOH through a process of bicarbonate buffering. Maintenance of this ratio in extra- and intracellular fluids is thought to be important for stabilizing macromolecular structure and function (*Reeves, 1972*; *Reeves, 1977*; *Somero, 1981*; *Truchot, 1973*). In vitro, shifts in hemolymph pH can alter the frequency and strength of the lobster cardiac rhythm (*Qadri et al., 2007*). Like hemolymph, intracellular pH generally decreases as temperature rises, but has been shown to change at varying rates in different tissues in the crab, *Callinectes sapidus* (*Wood and Cameron, 1985*). Active mechanisms for the maintenance of intracellular pH have been suggested in the crab, *Cancer pagurus* (*Golowasch and Deitmer, 1993*).

Both temperature and pH alter the biophysical parameters governing the activity of ion channels and pumps in excitable membranes. Under moderately acidic extracellular conditions, attenuation of sodium, calcium, and potassium currents by up to 60% has been shown in mammalian and nonmammalian systems. (*Courtney, 1979*; *Doering and McRory, 2007*; *Hille, 1968*; *Mozhaev et al., 1970*; *Tombaugh and Somjen, 1996*; *Wanke et al., 1979*; *Woodhull, 1973*; *Zhou and Jones, 1996*). The reversible effects of pH on these channels may result from protonation of an acidic group with $pK_a$ between 5.2 and 7.1 (*Tombaugh and Somjen, 1996*). Decreases in glutamatergic synaptic function (*Billups and Attwell, 1996*; *Bloch et al., 2001*; *Sinning et al., 2011*) and increases in GABAergic synaptic function (*Sinning and Hübner, 2013*) in response to extracellular acidification have also been shown.

Further, respiratory control in mammals is highly pH sensitive and important for the regulation of $CO_2$ and $O_2$ levels in the blood (*Richerson, 1998*) as $CO_2$ rapidly dissolves in and acidifies physiological solutions (*Nattie, 1999*). Multiple pH-sensitive ion channels and pumps, including a family of acid-sensing ion channels (ASICs) have been discovered in mammalian cells and are hypothesized to play a role in the maintenance of blood pH (*Buckler et al., 1991*; *Husson and Smith, 2018*; *Wang et al., 2018*).

Although the biophysical, ethological, and environmental implications of changing pH have been well studied, less is known about the effect of pH changes on neuronal circuits in marine invertebrates. Here, we study the effects of acute changes in extracellular pH on two well characterized neuronal circuits, the stomatogastric (STG) and cardiac (CG) ganglia, of the crab, *C. borealis*. These central pattern generators (CPGs) drive the coordinated and rhythmic muscle movements of the crab's stomach and heart, respectively. Although these CPGs are driven by numerically small neuronal circuits, their dynamics involve complex interactions between many intrinsic and synaptic currents.

Previous studies have shown that the pyloric rhythm of the STG and cardiac rhythm of the CG are remarkably robust to short- and long-term changes in temperature in both in vivo and ex vivo preparations (*Kushinsky et al., 2018*; *Marder et al., 2015*; *Soofi et al., 2014*; *Tang et al., 2010*; *Tang et al., 2012*). Despite similarly robust activity under moderate perturbation, these experiments have revealed animal-to-animal variability in network activity at extreme temperatures (*Haddad and Marder, 2018*; *Tang et al., 2010*; *Tang et al., 2012*). Given the relative insensitivity of the STG to moderate shifts in pH (*Golowasch and Deitmer, 1993*), we sought to probe the robustness and variability of these circuits to extreme pH perturbation. To provide insight into the animal-to-animal variability in response to this extreme perturbation, we challenged these preparations with saline whose pH was outside the range expected in the normal ocean environment. Both the pyloric and cardiac rhythms are remarkably robust to acute pH changes within the range tested. This is surprising given the sensitivity of many ion channels to pH in these ranges and suggests that networks can be more robust to pH changes than expected.

## Results

Two neuronal networks were studied in this paper. The stomatogastric ganglion (STG) of the crab, *Cancer borealis*, contains the neurons that generate two stomach rhythms, the fast pyloric rhythm and the slower gastric mill rhythm. The pyloric rhythm is driven by a three-neuron pacemaker kernel – one Anterior Burster (AB) and two Pyloric Dilator (PD) neurons. The Lateral Pyloric (LP) and Pyloric (PY) neurons fire out of phase with the PD neurons because they are rhythmically inhibited by the PD/AB group (*Marder and Bucher, 2007*). The cardiac ganglion (CG) generates the rhythm responsible for heart contraction, and consists of four pacemaker Small Cell (SC) neurons that drive five motor Large Cell (LC) neurons (*Cooke, 2002*).

A schematic diagram of the stomatogastric nervous system preparation is found in *Figure 1A*. Intracellular recordings were made from the somata of the desheathed STG and examples of the LP and PD neuron waveforms are shown. Extracellular recordings from the motor nerves are indicated. An example of the triphasic activity of the LP, PD, and PY neurons is seen in the inset trace next to the lateral ventricular nerve (*lvn*). The connectivity diagram of the major neuron classes of the triphasic pyloric rhythm is given in *Figure 1B*.

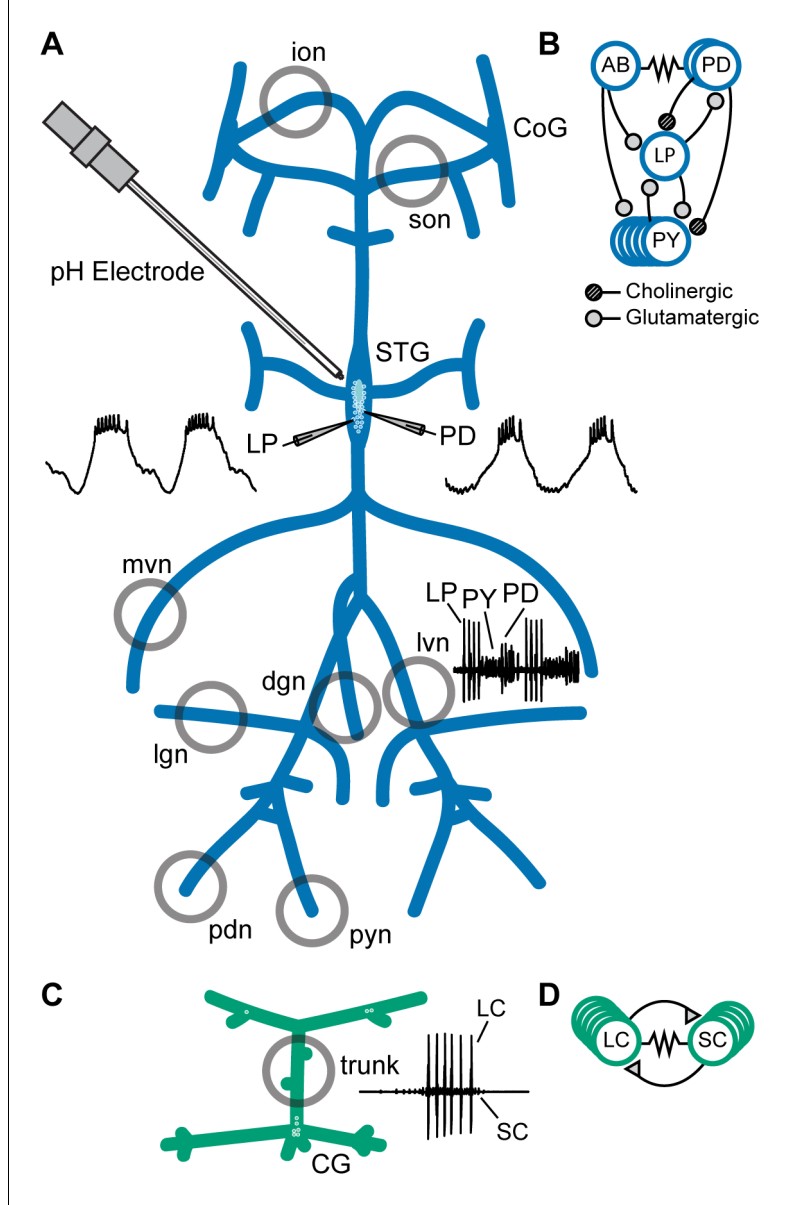

**Figure 1.** Preparations and circuit diagrams. (**A**) Schematic of the stomatogastric nervous system preparation. Extracellular electrodes were placed in vaseline wells (gray circles) drawn around nerves of interest. An example extracellular nerve recording from the lateral ventricular nerve (*lvn*) shows two cycles of the triphasic pyloric rhythm containing spikes from the Lateral Pyloric (LP), Pyloric (PY), and Pyloric Dilator (PD) neurons. Example intracellular recordings from the LP and PD neurons are displayed. (**B**) Simplified diagram of the pyloric circuit. Filled circles represent inhibitory chemical synapses; resistor symbol represents electrical coupling. (**C**) Schematic of the cardiac ganglion preparation. Extracellular electrodes were placed in a well (gray circle) around the trunk of the preparation. An example extracellular recording shows one burst of the Small Cell (SC) and Large Cell (LC) neurons. (**D**) Diagram of the cardiac circuit. Filled triangles represent excitatory chemical synapses; the resistor symbol represents electrical coupling.

DOI: https://doi.org/10.7554/eLife.41877.002

A schematic diagram of the crab cardiac ganglion preparation shows an example of one burst of SC and LC activity recorded from the trunk (*Figure 1C*). *Figure 1D* shows the connectivity diagram of the cardiac ganglion.

## The pyloric rhythm is surprisingly robust to extreme changes in pH

To characterize the response of the pyloric rhythm to acute changes in pH, superfused saline was exchanged every 15 min in steps of approximately pH 0.5 from a control pH of 7.8 to an extreme pH of either 5.5 or 10.4. Following the first acid or base step protocol, preparations were allowed to recover for a minimum of 30 min at control pH until the frequency of the pyloric rhythm approached that of controls. Preparations were then subjected to a step protocol in the opposite direction followed by a second recovery period. Acid- or base-first protocols were counterbalanced.

Recordings and analysis from an example STG experiment with an acid-first protocol are shown in *Figure 2*. Each box contains simultaneous intracellular recordings of the PD and LP neurons and extracellular recordings of the *lvn* during the last minute at each pH step (*Figure 2A*). STG #1 demonstrated a normal triphasic rhythm in control saline at pH 7.8 (*Figure 2A*; top left). As the preparation was subjected to more acidic saline, the rhythm remained triphasic until the most acidic pH, 5.5. Control activity was recovered when the preparation was once again placed in control saline (*Figure 2A*; top right). The bottom row shows the same preparation in basic conditions where it

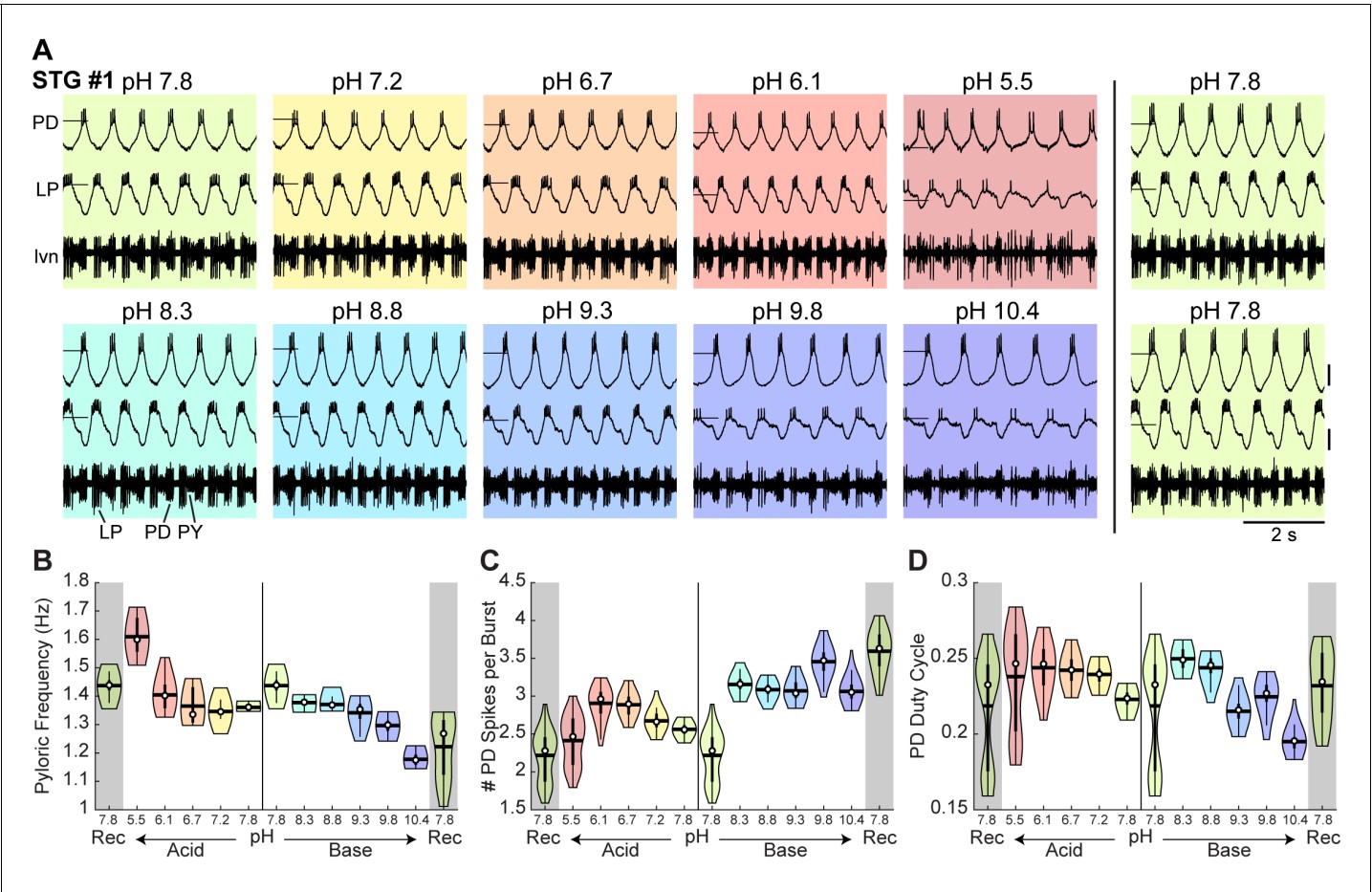

**Figure 2.** Robust pyloric rhythm activity across pH. (A) Example recordings from a stomatogastric ganglion experiment with an acid-first protocol. Intracellular recordings of the PD and LP neurons and extracellular recordings of the *lvn* are shown. Each colored box displays 5 s of recordings taken from the last minute at each pH step. The experiment can be read left to right then top to bottom in chronological order. Horizontal lines indicate a reference membrane potential of −40 mV; vertical lines indicate a scale of 10 mV. (B) Pyloric frequency, (C) number of PD spikes per burst, and (D) PD duty cycle were calculated for the last 8 min of each pH step. Violin plots show the KDE distribution, mean, median, IQR, and 95% CI for each measure across pH conditions. Recoveries from acid and base are displayed in the shaded gray regions on the far ends of each plot.

DOI: https://doi.org/10.7554/eLife.41877.003

The following source data is available for figure 2:

**Source data 1.** Numerical data represented in panel B, C, and D of *Figure 2*.
DOI: https://doi.org/10.7554/eLife.41877.004

remained triphasic, although with fewer spikes per burst and lower amplitude slow waves at pH 10.4. Again, the preparation recovered a canonical triphasic rhythm in control saline as seen in the bottom right.

Measures of the pyloric rhythm burst frequency, PD spikes per burst, and PD duty cycle (the fraction of the pyloric rhythm's period during which the PD neurons were active) were calculated for the last 8 min of each 15 min pH step. Violin plots reveal the distribution of these measures for STG #1 at each pH (*Figure 2B–D*). The pyloric burst frequency of STG #1 increased in acid and decreased in base (*Figure 2B*). The number of PD spikes per burst decreased at pH 5.5 (*Figure 2C*). The duty cycle of the PD neurons in STG #1 decreased slightly in base (*Figure 2D*).

While many preparations maintained surprisingly robust pyloric rhythms from pH 5.5 to 10.4, others exhibited disrupted patterns of activity at the most extreme pH conditions. The activity of two additional STG preparations across the range of pH tested is shown in *Figure 3A*. Both preparations displayed robust activity across a nearly 125-fold range of hydrogen ion concentration but became weakly active or silent at pH 5.5. At pH 10.4, STG #2 was slow and weakly triphasic while STG #3 retained a strong triphasic rhythm. These examples highlight both the animal-to-animal variability of the pyloric rhythm at control conditions and the variable effects of extreme acidosis or alkylosis on this network.

To characterize these effects across all preparations, we defined five states of activity: (1) 'normal triphasic' rhythm containing PD, LP, and PY with a minimum of three spikes per burst for each unit; (2) 'weak triphasic' rhythm retaining all three units with some units spiking only once or twice per cycle; (3) 'intermittent triphasic' rhythm describing rhythmic activity with only some units active; (4) 'all silent'; and (5) 'atypical activity' or activity that could not be categorized under the first four definitions (*Figure 3B*). Preparations were categorized systematically according to the criteria outlined in Materials and methods. The mean fraction of time that all preparations spent in these five states during the last 8 min of each pH step was analyzed (*Figure 3C*).

Rhythms rarely transitioned intermittently between two states. For example, once a pyloric rhythm had transitioned from normal triphasic to weak triphasic, it rarely transitioned back to normal triphasic until after recovery in control pH. Further, rhythms generally transitioned in a stereotypical pattern. The pyloric rhythm often transitioned from normal triphasic to weak triphasic to intermittent triphasic to all silent. During recovery, this transition pattern was reversed. Both acid and base significantly decreased the fraction of time that preparations were rhythmic, a combined metric of states 1 (normal triphasic) and 2 (weak triphasic) (*Figure 3—figure supplement 1*). Preparations were significantly less triphasically rhythmic at pH 5.5 and pH 10.4 compared to control pH 7.8.

To describe these effects quantitatively, measures of the pyloric rhythm frequency, the number of PD spikes per burst, and PD duty cycle were calculated. Violin plots give pooled distributions for each pH across all preparations (*Figure 3D–F*). Mean pyloric burst frequency was relatively invariant across pH values in the presence of acid, but varied significantly across base steps (*Figure 3D*). Pyloric burst frequency at pH 9.3 and 10.4 was significantly slower than that at control. Both acid and base significantly affected the mean number of PD spikes per burst (*Figure 3E*). The number of spikes per burst was significantly reduced at pH 5.5. Additionally, there was a significant effect of acid and base on the mean PD duty cycle (*Figure 3F*). Paired samples t-tests revealed a slight increase in mean PD duty cycle from control pH 7.8 to pH 6.1.

The pooled distributions for these three measures were highly variable for all pH conditions reflecting the animal-to-animal variability in the pyloric rhythm. We plotted the distributions for all 15 STG preparations at control pH 7.8 and found similarly variable activity at baseline conditions (*Figure 3—figure supplement 2*).

## Semi-isolated pyloric neurons are sensitive to extreme pH

To determine how the intrinsic properties of neurons respond to pH, we analyzed several characteristics of the intracellular recordings from the PD and LP neurons (*Figure 4*). We isolated the neurons from most of their pyloric network synaptic inputs by blocking the glutamatergic inhibitory synapses with $10^{-5}$ M picrotoxin (PTX) (*Marder and Eisen, 1984*) (*Figure 1B*). We analyzed mean resting membrane potential (mV), spike amplitude (mV), and burst or spiking frequencies (Hz) for both cells in the presence of PTX as a function of pH. The waveforms of the PD and LP neurons from an example preparation are shown prior to PTX superfusion (*Figure 4A*; leftmost traces). Note the large LP-evoked inhibitory post-synaptic potentials (IPSPs) in the trough of the PD neuron waveform and the

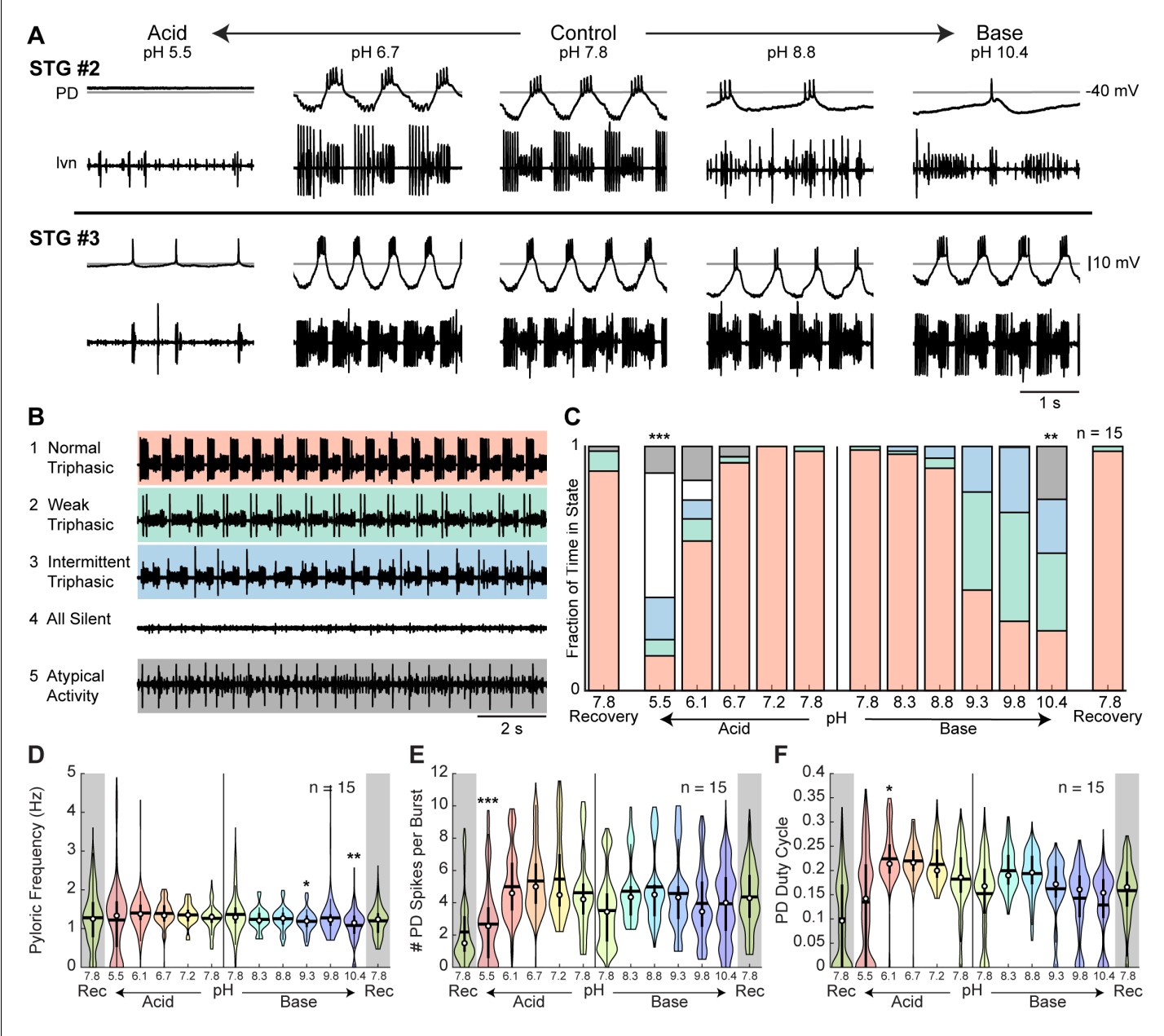

**Figure 3.** Variability of pyloric rhythm activity at extreme pH. (**A**) Two additional stomatogastric ganglion experiments displaying 3 s of intracellular PD and extracellular *lvn* recordings. Horizontal lines indicate a reference membrane potential of −40 mV; vertical line indicates a scale of 10 mV. (**B**) Five states were defined to characterize pyloric rhythm activity. Examples of activity for each state are given. (**C**) Stacked bars give the mean fraction of time that all 15 preparations spent in each state. (**D**) Pyloric rhythm frequency, (**E**) number of PD spikes per burst, and (**F**) PD duty cycle were calculated and pooled across all STG preparations for each pH step. Violin plots show the KDE distribution, mean, median, IQR, and 95% CI for each measure across pH conditions. Recoveries from acid and base are displayed in the shaded gray regions on the far ends of each plot. Asterisks denote statistical significance revealed by paired samples t-tests with Bonferroni correction (*p<0.05; **p<0.01; ***p<0.001).

DOI: https://doi.org/10.7554/eLife.41877.005

The following source data and figure supplements are available for figure 3:

**Source data 1.** Numerical data represented in panel C, D, E, and F of *Figure 3* and and panel A, B, and C of *Figure 3—figure supplement 2*.
DOI: https://doi.org/10.7554/eLife.41877.008

**Figure supplement 1.** Statistical analysis of the effects of pH on the pyloric rhythm.
DOI: https://doi.org/10.7554/eLife.41877.006

**Figure supplement 2.** Variability of pyloric rhythm activity at control pH.
DOI: https://doi.org/10.7554/eLife.41877.007

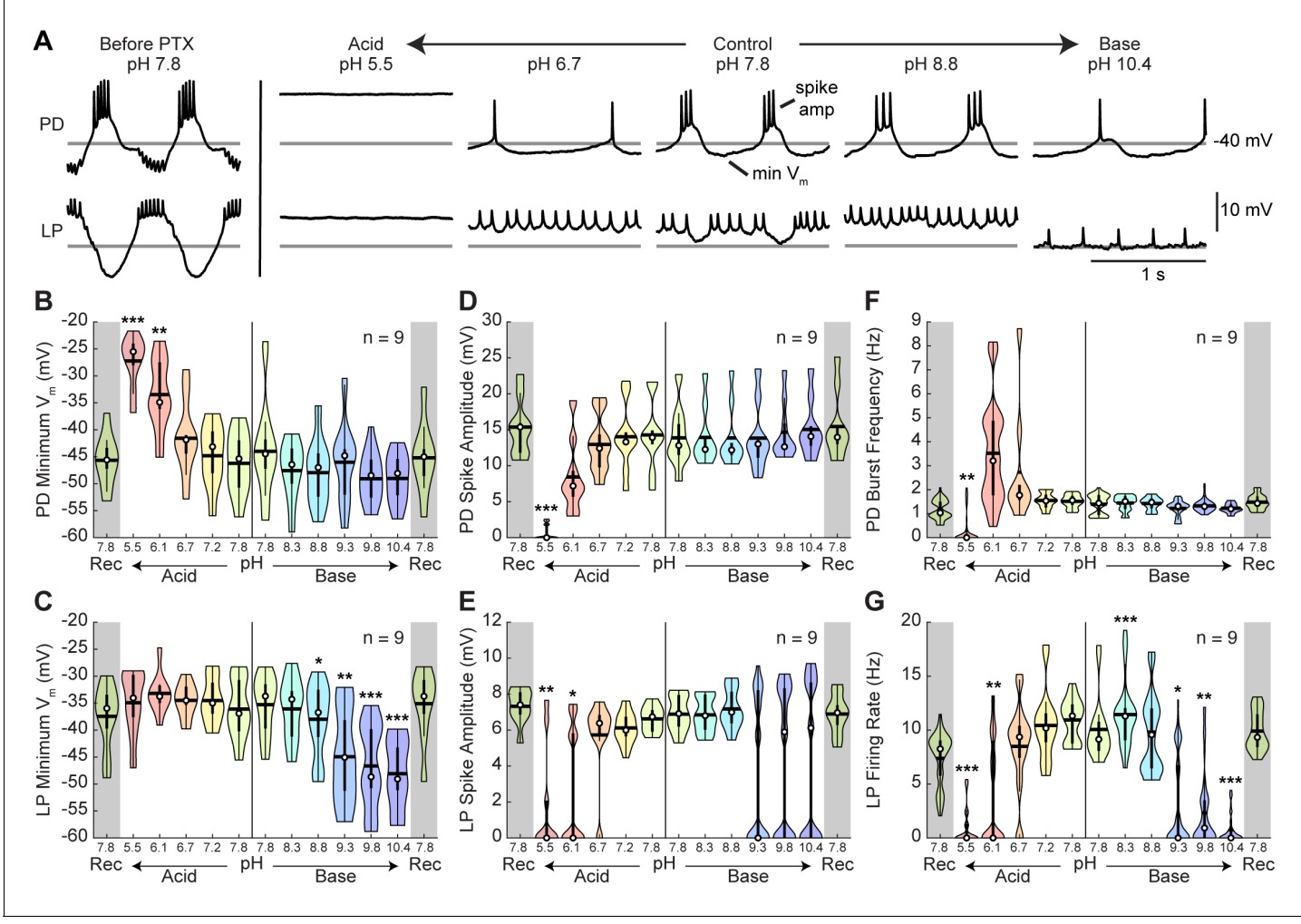

**Figure 4.** Intracellular characteristics of semi-isolated pyloric neurons, PD and LP. Several characteristics of the PD and LP neurons in the presence of picrotoxin (PTX) were measured for the last minute of each pH condition. (A) Example intracellular recordings of PD and LP neurons prior to PTX application and in the presence of PTX across pH conditions. Horizontal lines indicate a reference membrane potential of −40 mV; the vertical line indicates a scale of 10 mV. (B,C) Minimum membrane potential and (D,E) spike amplitude are plotted for LP and PD as a function of pH. (F) PD burst frequency and (G) LP firing rate are also plotted at each pH. Violin plots show the KDE distribution, mean, median, IQR, and 95% CI for each measure across pH conditions. Recoveries from acid and base are displayed in the shaded gray regions on the far ends of each plot. Asterisks denote statistical significance revealed by paired samples t-tests with Bonferroni correction (*p<0.05; **p<0.01; ***p<0.001).
DOI: https://doi.org/10.7554/eLife.41877.009

The following source data and figure supplement are available for figure 4:

**Source data 1.** Numerical data represented in panel B, C, D, E, F, and G of *Figure 4*.
DOI: https://doi.org/10.7554/eLife.41877.011

**Figure supplement 1.** Statistical analysis of the effects of pH on semi-isolated PD and LP neurons.
DOI: https://doi.org/10.7554/eLife.41877.010

large amplitude inhibition of the LP caused by activity of the PD, AB, and PY neurons. Following application of PTX at control pH 7.8 (*Figure 4A*; center traces), the PD neuron was still bursting but the LP-evoked IPSPs were entirely blocked. Most of the inhibitory inputs to the LP neuron were blocked, leaving only the cholinergic inhibition contributed by the PD neurons. At pH 6.7, the PD neuron of this preparation lost most of its slow wave activity. It then became silent and depolarized in pH 5.5 saline. At pH 8.8, the PD burst was largely intact, and at pH 10.4, the neuron showed single spike bursts. The LP neuron fired tonically from pH 6.7 to 10.4, again showing loss of activity at pH 5.5 similar to PD.

Violin plots show pooled values for the most hyperpolarized levels (minimum voltages) of the membrane potential for PD and LP neurons (*Figure 4B,C*). For moderate shifts in pH, the membrane potential was fairly stable. At extreme acid, the mean PD neuron membrane potential depolarized significantly, while the mean LP neuron membrane potential remained relatively constant (*Figure 4— figure supplement 1*). In contrast, the PD neuron's membrane potential in basic saline was relatively constant, even at extreme base, but the LP neuron's membrane potential significantly hyperpolarized. The depolarization of PD at pH 5.5 and 6.1 and the hyperpolarization of LP at pH 8.8, 9.3, 9.8, and 10.4 were significantly different from control saline.

Additionally, there was a slight effect of pH on mean spike amplitude at the most extreme pH conditions for both the LP and PD neurons (*Figure 4D,E*). Acid significantly affected both PD and LP neuron spike amplitude while alkylosis had an effect on LP, but not PD neurons (*Figure 4—figure supplement 1*). At pH 5.5, spike amplitude was significantly attenuated for both LP and PD. Additionally, LP spike amplitude was significantly decreased at pH 6.1 while PD was only moderately affected.

There was a significant effect of both acidic and basic saline on mean PD burst frequency and LP firing rate (*Figure 4F,G*; *Figure 4—figure supplement 1*). Mean PD burst frequency was significantly decreased at pH 5.5. The LP firing rate was significantly reduced in pH 5.5, 6.1, 8.3, 9.3, 9.8, and 10.4 compared to control pH 7.8.

## Rhythmic gastric-like activity was elicited upon exposure to and recovery from extreme acid and base

The STG contains a second slower central pattern generating circuit known as the gastric mill rhythm (*Marder and Bucher, 2007*). Unlike the pyloric rhythm which contains a pacemaker kernel, the gastric rhythm is controlled by the reciprocal alternation of activity driven by descending neuromodulatory inputs (*Marder and Bucher, 2007*; *Nusbaum et al., 2017*). The principal neurons involved in the gastric mill rhythm are the Lateral Gastric (LG), Medial Gastric (MG), Lateral Posterior Gastric (LPG), Gastric Mill (GM), Dorsal Gastric (DG), and Interneuron 1 (Int1) neurons (*Mulloney and Selverston, 1974a*; *Mulloney and Selverston, 1974b*).

The gastric mill rhythm is often silent in dissected STG preparations and requires stimulation of descending and/or sensory neurons to elicit activity (*Blitz et al., 1999*). Interestingly, in 10 of 15 preparations a gastric-like rhythm appeared at pH 8.8 or above, and 4 of 15 showed this type of activity in acid at or below pH 6.1. Further, a strong gastric-like rhythm was elicited upon recovery from extreme acid in 5 of 15 preparations and from extreme base in 7 of 15. Preparations in which gastric rhythms were seen in one of these conditions were likely to display gastric-like activity in the other conditions.

An example preparation in control pH 7.8 saline is shown as it recovers activity after exposure to pH 5.5 saline (*Figure 5*). Intracellular recordings from the LP and PD neurons and extracellular recordings from five nerves – lateral ventricular (*lvn*), medial ventricular (*mvn*), dorsal gastric (*dgn*), lateral gastric (*lgn*), and inferior oesophageal (*ion*) – are shown. In addition to axons of the LP, PD, and PY neurons, the *lvn* contains the LG axon. The *mvn* contains axons from two neurons, the Inferior Cardiac (IC) and the Ventricular Dilator (VD). Inhibition of IC and VD was coincident with LG bursting. The *dgn* shows GM and DG activity and the *lgn* contains LG activity. Recordings from the *ion* reveal Modulatory Commissural Neuron 1 (MCN1) activity. One period of the gastric mill rhythm can be defined by the time from the onset of one LG burst to the next.

Over the 20 min shown, there was a clear increase in both gastric and pyloric activity with the LP and PD neurons becoming rhythmic and the emergence of strong rhythmic activity of the MCN1, LG, DG, and GM neurons (*Figure 5A*). At the beginning of this recovery period, both the PD and LP neurons were silent, reflecting loss of activity in pH 5.5 (*Figure 5B*). Strong LG neuron bursts were timed with hyperpolarizations of the LP neuron. This is consistent with previous findings that the neurons driving the gastric mill rhythm synapse onto the pyloric network and that gastric mill activity correlates with slowing of the pyloric rhythm (*Bucher et al., 2007*). A few minutes later, the LP and PD neurons started to recover rhythmic slow waves (*Figure 5C*). The LP and the second PD neuron – seen here on the *lvn* recording – were bursting. Both neurons became silent due to a strong inhibitory input coinciding with strong LG and MCN1 activity. Shortly thereafter, the LP and PD neurons were firing rhythmically (*Figure 5D*). Depolarizing inhibition resulted in tonic firing of LP and no activity on the intracellular recording of PD. Finally, the LP and PD neurons were bursting

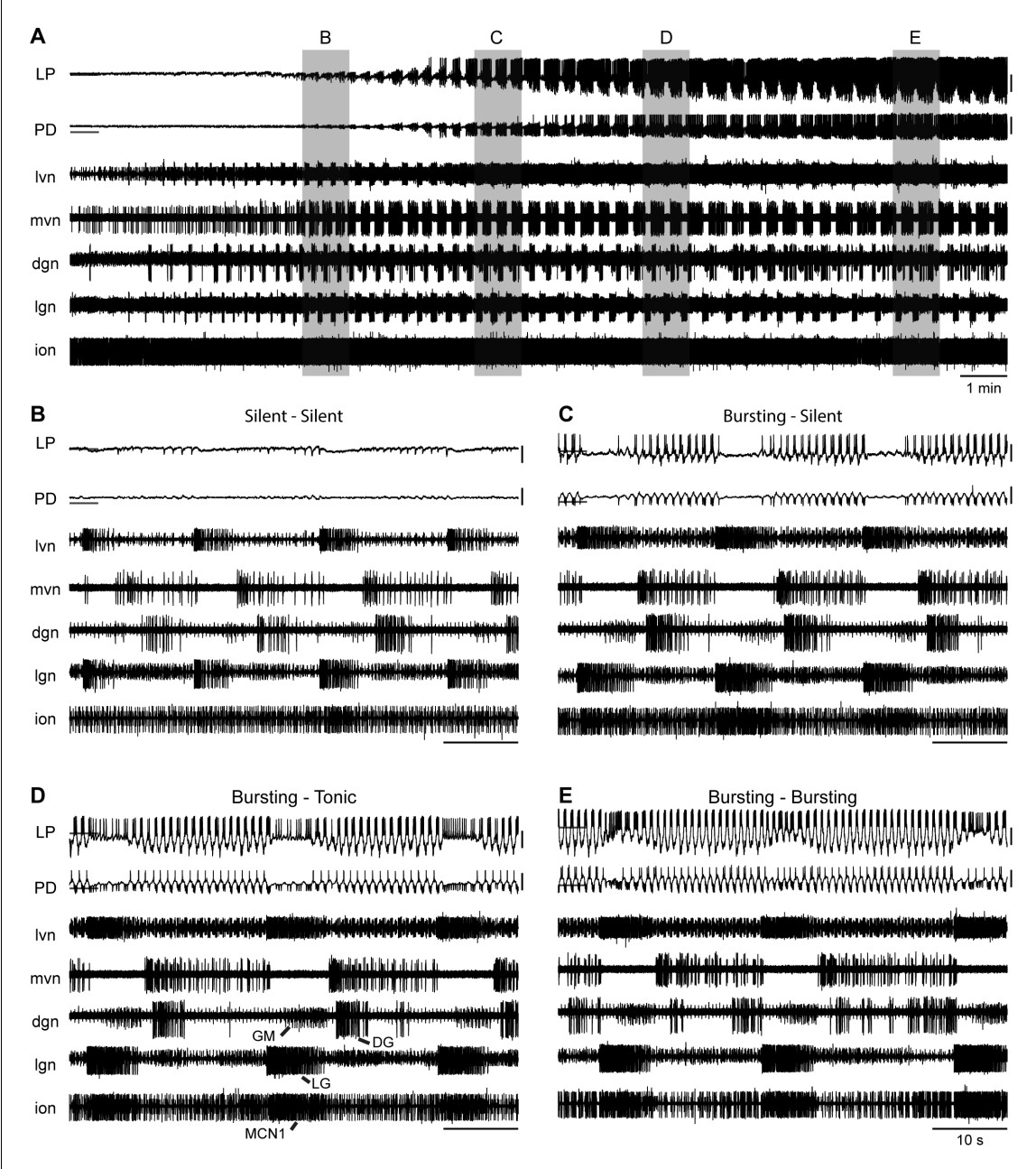

**Figure 5.** Rhythmic gastric-like activity upon recovery from extreme acid. (**A**) 20 min of recording are shown from an example experiment where the ganglion had become silent at pH 5.5 and began recovering rhythmic activity in control pH 7.8 saline. Intracellular recordings from LP and PD neurons and extracellular recordings from five nerves – *lvn*, *mvn*, *dgn*, *lgn*, and *ion* – are displayed. Horizontal lines indicate a reference membrane potential of −40 mV; vertical lines indicate a scale of 10 mV. Gray boxes correspond to the one-minute snapshots enlarged in subsequent panels respective to time. (**B–E**) Titles describe the pyloric neuron activity during and between LG bursts.

DOI: https://doi.org/10.7554/eLife.41877.012

(*Figure 5E*). Inhibitory input coincident with LG and MCN1 activity resulted in depolarization of the PD and LP neurons and an increased duty cycle of LP bursting.

The rhythmic gastric-like activity seen here is similar to gastric mill rhythms elicited upon stimulation of the Ventral Cardiac Neurons (VCNs) (*Beenhakker et al., 2007*; *Saideman et al., 2007*; *White and Nusbaum, 2011*). Studies have shown that stimulation of the VCNs triggers activation of MCN1 and Commissural Projection Neuron 2 (CPN2) in the commissural ganglia (CoGs). This

MCN1/CPN2 gastric mill rhythm drives the alternation of activity of the protractor motor neurons – LG, GM, MG, and IC – and the retractor neurons – DG, Int1, and VD. We see similar activity here with strong MCN1 bursts on the *ion* corresponding to strong LG and GM bursts on the *lgn* and *dgn*, respectively, in alternation with DG bursts on the *dgn*.

## The cardiac rhythm is robust to acute changes in pH

To characterize the response of the cardiac rhythm to pH, we bath superfused cardiac ganglion preparations with saline between pH 5.5 and pH 10.4 using the same protocol described above for stomatogastric ganglion preparations. Example extracellular recordings are shown from the cardiac ganglia of two animals during the last minute of each pH step (*Figure 6A*). As shown in the top row of CG #1, the ganglion started in control saline at pH 7.8 and demonstrated a normal rhythm of Small and Large Cells bursting together. As the preparation was subjected to both acidic and basic saline, the rhythm remained. In contrast, the cardiac rhythm in CG #2 became less rhythmic in pH

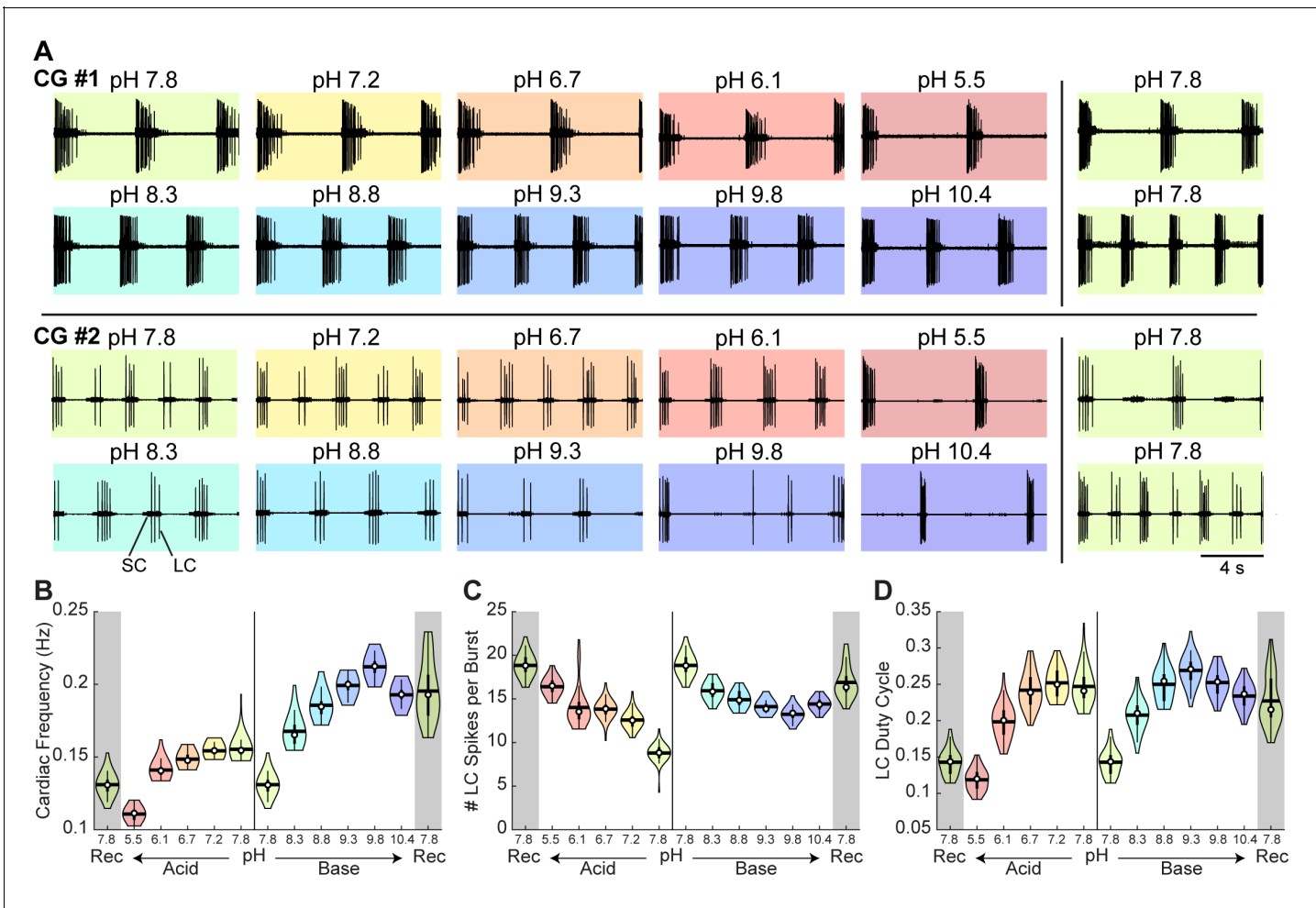

**Figure 6.** Robust and variable cardiac rhythm activity across pH. (**A**) Two example cardiac ganglion experiments with an acid-first protocol. Each colored box displays 12 s of extracellular recordings of the trunk taken from the last minute of each pH condition. Small Cell (SC) and Large Cell (LC) activity is visible. Each experiment can be read left to right then top to bottom in chronological order. (**B**) Cardiac frequency, (**C**) number of LC spikes per burst, and (**D**) LC duty cycle were calculated for CG #1 for each pH step. Violin plots show the KDE distribution, mean, median, IQR, and 95% CI for each measure across pH conditions. Recoveries from acid and base are displayed in the shaded gray regions on the far ends of each plot.

DOI: https://doi.org/10.7554/eLife.41877.013

The following source data is available for figure 6:

**Source data 1.** Numerical data represented in panel B, C, and D of *Figure 6*.
DOI: https://doi.org/10.7554/eLife.41877.014

5.5 and in pH 9.8 and above. A normal bursting rhythm recovered after superfusion of control saline as seen in the bottom right.

Measures of cardiac ganglion rhythm frequency, LC spikes per burst, and LC duty cycle were calculated for CG #1. Violin plots reveal the distribution of these measures at each pH (*Figure 6B–D*). The cardiac frequency of CG #1 decreased in acid and increased in base (*Figure 6B*). Further, the number of LC spikes per burst increased in acid (*Figure 6C*) while the LC duty cycle for CG #1 decreased slightly in acid and base (*Figure 6D*). Similar to STG #1, CG #1 retained robust activity throughout the entire range of pH tested.

To characterize these effects, we defined four states of activity: (1) 'SC and LC bursting' rhythm containing both units with a minimum of one LC spike per SC burst; (2) 'SC bursting only' rhythm containing only SC bursts with no or inconsistent LC spiking; (3) 'all silent'; and (4) 'atypical activity' that could not be categorized under the first three definitions (*Figure 7A*). The mean fraction of time that all preparations spent in these states is plotted (*Figure 7B*). The cardiac rhythm usually transitioned from SC and LC bursting to SC bursting only to all silent. This transition pattern was reversed during recovery. The mean fraction of time that preparations were rhythmic (state 1 – SC and LC bursting) was significantly affected by both acid and base (*Figure 7—figure supplement 1*). Rhythmic activity was significantly decreased at pH 5.5, pH 9.3, pH 9.8, and pH 10.4 compared to control pH 7.8.

To describe these effects quantitatively, measures of rhythm frequency, the number of LC spikes per burst, and LC duty cycle were calculated and their distributions are displayed in violin plots (*Figure 7C–E*). Cardiac rhythm frequency declined in both acidic and basic saline (*Figure 7C*; *Figure 7—figure supplement 1*). At pH 10.4, the cardiac rhythm was significantly slower. The mean number of LC spikes per burst was significantly affected in both acid and base (*Figure 7D*). The number of LC spikes per burst was significantly increased at pH 5.5 and pH 6.1 and decreased at pH 9.8 and pH 10.4. There was a significant effect of base but not acid on mean LC duty cycle (*Figure 7E*).

Similar to the STG, we observed a large spread in pooled measures across all pH conditions, reflecting the animal-to-animal variability in these networks. We plotted the distributions for all CG preparations at baseline and noted highly variable cardiac rhythm activity independent of the pH perturbation (*Figure 7—figure supplement 2*).

## The cardiac and pyloric rhythms are differentially sensitive to pH

To compare the effect of pH on the cardiac and pyloric rhythms, the distributions of the fraction of time that each preparation retained a normal rhythm were compared (*Figure 8A*). A normal rhythm was defined as a triphasic rhythm (a combined metric of states 1 and 2) and Small Cells and Large Cells bursting together (state 1) for the pyloric and cardiac rhythms, respectively. A comparison between the rhythmicity of these two ganglia across pH reveals similar distributions with maxima around control pH 7.8 and minima at extreme pH values. Interestingly, these distributions are asymmetrical, as the CG was more sensitive to extreme base whereas the STG was more sensitive to extreme acid. There were significant main effects of pH and ganglion as well as an interaction between pH and ganglion on rhythmicity in both acidic and basic solutions (*Figure 8—figure supplement 1*). The pyloric rhythm was significantly less rhythmic at pH 5.5, but significantly more rhythmic at pH 9.3 and 9.8 compared to the cardiac rhythm.

To understand better the amount of animal-to-animal variability in these two rhythms, the activity of individual preparations was plotted in control pH 7.8, extreme acid pH 5.5, and extreme base pH 10.4 (*Figure 8B*). All preparations were rhythmically bursting at control pH – indicated by darkly colored boxes – and became less rhythmic – lighter colored – in the presence of extreme acid. However, 14 of 15 STG preparations ceased firing after 15 min of exposure to pH 5.5. In contrast, only 6 of 15 CG preparations showed reductions in rhythmic activity at pH 5.5. Three of 15 CG preparations maintained a normal rhythm in every pH condition. Interestingly, STG preparations that showed decrements in activity during basic conditions were extremely susceptible to reductions in activity during extreme acid. The opposite is true in CG preparations suggesting that activity in base is a better predictor of acid activity in the STG and vice versa in the CG. This finding also suggests that some preparations were more susceptible to the effects of pH than others.

Furthermore, strong correlations are found between baseline frequency and frequency in extreme acid or base for PD neurons. Strong correlations are also found between baseline LC frequency and

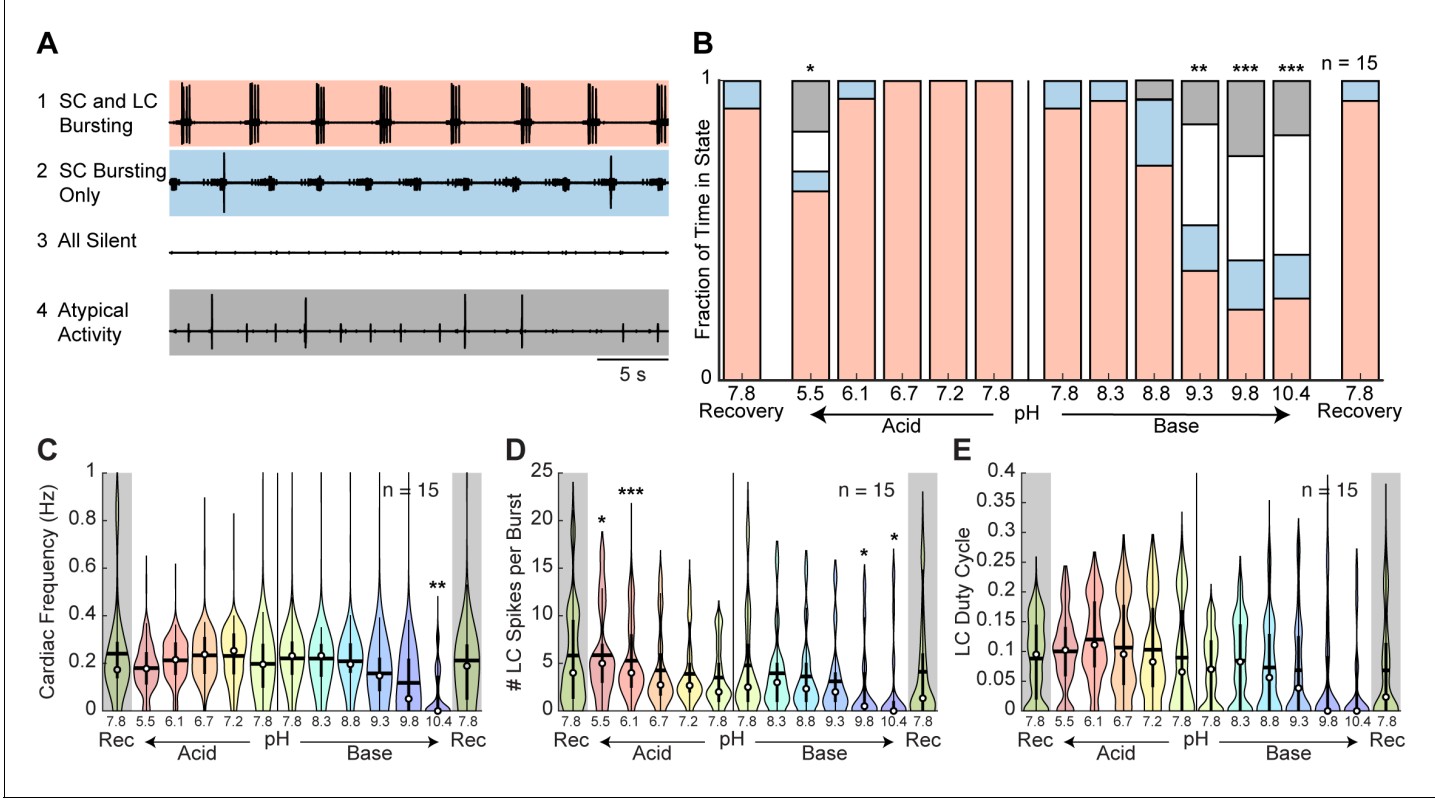

**Figure 7.** Characteristics of cardiac rhythm activity across pH. (**A**) Four states were defined to characterize cardiac rhythm activity. Examples of activity for each state are given. (**B**) Stacked bars give the mean fraction of time that all 15 preparations spent in each state for each pH step. (**C**) Cardiac rhythm frequency, (**D**) number of LC spikes per burst, and (**E**) LC duty cycle were calculated and pooled across all CG preparations for each pH step. Violin plots show the KDE distribution, mean, median, IQR, and 95% CI for each measure across pH conditions. Recoveries from acid and base are displayed in the shaded gray regions on the far ends of each plot. Asterisks denote statistical significance revealed by paired samples t-tests with Bonferroni correction (*p<0.05; **p<0.01; ***p<0.001).

DOI: https://doi.org/10.7554/eLife.41877.015

The following source data and figure supplements are available for figure 7:

**Source data 1.** Numerical data represented in panel B, C, D, and E of *Figure 7* and panel A, B, and C of *Figure 7—figure supplement 2*.
DOI: https://doi.org/10.7554/eLife.41877.018
**Figure supplement 1.** Statistical analysis of the effects of pH on the cardiac rhythm.
DOI: https://doi.org/10.7554/eLife.41877.016
**Figure supplement 2.** Variability of cardiac rhythm activity at control pH.
DOI: https://doi.org/10.7554/eLife.41877.017

LC frequency in extreme acid, but not in extreme base. In other words, the preparations that are weaker or slower at control conditions remain slower at extreme pH conditions. This suggests that the variability seen here is inherent in the population and is likely not caused by the pH perturbation.

## Discussion

Circuit dynamics depend on the properties of the constituent neurons and their synaptic connections. Likewise, the intrinsic excitability of an individual neuron depends on the number and properties of its voltage- and time-dependent channels. Crab central pattern generating circuits are robust and adaptable to a large range of temperatures (*DFO, 2009*; *Haddad and Marder, 2018*; *Rinberg et al., 2013*; *Tang et al., 2010*; *Tang et al., 2012*). Previous research revealed robust activity and increasing frequency of the pyloric and cardiac rhythms in response to increasing temperature in both in vivo and ex vivo preparations (*Kushinsky et al., 2018*; *Tang et al., 2010*; *Tang et al.,*

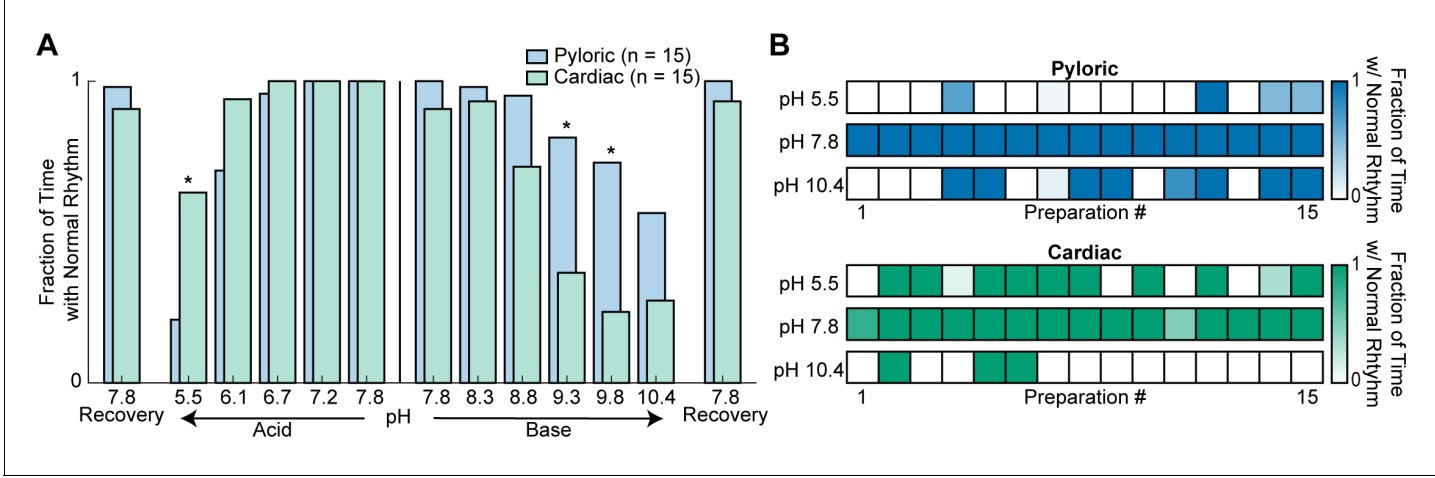

**Figure 8.** Rhythmicity of the cardiac and pyloric rhythms compared across pH. (**A**) Mean fraction of time that both the pyloric (blue) and cardiac (green) rhythms displayed normal activity is plotted as a function of pH. Differences between the activity of the two rhythms were analyzed by independent samples t-tests at each pH. Recovery from acid and base are displayed on the far ends of the plot. Asterisks denote statistical significance with Bonferroni correction (*p<0.05; **p<0.01; ***p<0.001). (**B**) Rhythmicity of individual animal preparations is plotted for extreme acid (pH 5.5), control (pH 7.8) and extreme base (pH 10.4) saline conditions. Each column of boxes represents a single preparation, with position across conditions remaining constant. The saturation of each box represents the mean fraction of time with a normal rhythm as indicated by the color bars on the right.
DOI: https://doi.org/10.7554/eLife.41877.019

The following source data and figure supplement are available for figure 8:

**Source data 1.** Numerical data represented in panel A and B of *Figure 8*.
DOI: https://doi.org/10.7554/eLife.41877.021
**Figure supplement 1.** Statistical analysis of the differential effects of pH on the pyloric and cardiac rhythms.
DOI: https://doi.org/10.7554/eLife.41877.020

*2012*). Contrastingly, increasing pH reveals non-linear effects on activity, which may suggest more complex mechanisms.

Given that the physiological properties of ion channels are sensitive to pH (*Anwar et al., 2017*; *Bayliss et al., 2015*; *Catterall, 2000*; *Cens et al., 2011*; *Cook et al., 1984*; *Doering and McRory, 2007*; *Guarina et al., 2017*; *Harms et al., 2017*; *Hille, 1968*; *Mahapatra et al., 2011*; *Marcantoni et al., 2010*; *Tombaugh and Somjen, 1996*; *Vilin et al., 2012*; *Zhou et al., 2018*), one might imagine that a neuronal circuit might be as sensitive to changes in pH as its most sensitive ion channels. Therefore, it is surprising that the central pattern generating rhythms of both the STG and the CG of the crab, *C. borealis*, are relatively insensitive to acute pH change from about pH 6.1 to pH 8.8 while the individual functions of many ion channels are known to be considerably altered within this range.

One possible explanation for these results is that crustacean ion channels are more robust to pH change than channels from other species. In most vertebrate animals, pH is carefully regulated. Slight acidosis or alkalosis can have deleterious effects on many aspects of vertebrate physiology (*Chesler, 2003*; *Fencl et al., 1966*; *Nattie, 1999*), which may be partially a consequence of the relative sensitivity of many vertebrate ion channels and synapses to pH. Unfortunately, little is known about the pH sensitivity of crustacean ion channels, but it would be surprising if it differed drastically from that seen in other animals as there is considerable homology across phylogeny in channel structure and function. Recent evidence suggests that acid-sensing ion channels are evolutionarily conserved, which presents the possibility of specialized channels for the modulation of pH in crustacean ganglia (*Lynagh et al., 2018*). However, the possibility remains that modest evolutionary changes in channel structure occurred to allow endothermic animals to function in high temperature and low pH conditions. Evidence of evolved pH robustness has been shown in mammals where smaller voltage-gated $Na^+$ and ASIC-mediated currents likely confer greater resistance to acid-induced cell death in the naked mole-rat brain compared to that of the mouse (*Husson and Smith, 2018*).

Another possibility for the circuit robustness seen in this study could be that there are compensatory and/or correlated changes in the effects of pH across the population of channels in these networks. Therefore, one prediction of the relative pH insensitivity of these networks is that numerous pH sensitive changes occur across the population of ion channels, but that these circuits have evolved sets of correlated ion channels that compensate for these changes (*O'Leary and Marder, 2016*; *O'Leary et al., 2013*; *O'Leary et al., 2014*; *Temporal et al., 2012*; *Temporal et al., 2014*; *Tobin et al., 2009*). Consistent with this possibility is the finding that in mammalian thalamocortical neurons, the counterbalancing activities of HCN and TASK channels restricts the effect of extracellular acidification (pH 7.3 to 6.3) on resting membrane potential (*Meuth et al., 2006*).

In addition to the relative pH insensitivity of these circuits, we were surprised that the cardiac and pyloric rhythms of *C. borealis* are differentially sensitive to acid and base. This was unexpected as there is almost a complete overlap of ion channel genes expressed in the two ganglia (*Northcutt et al., 2016*; *Ransdell et al., 2013a*; *Ransdell et al., 2013b*; *Schulz et al., 2006*; *Schulz et al., 2007*; *Tobin et al., 2009*). This differential sensitivity could be due to differences in the burst generating mechanisms of the two networks. The pyloric rhythm depends heavily on a bursting pacemaker neuron and on post-inhibitory rebound as a timing mechanism (*Harris-Warrick et al., 1995a*; *Harris-Warrick et al., 1995b*; *Hartline and Gassie, 1979*). In contrast, the cardiac ganglion depends on strong excitatory drive from the pacemaker neurons (*Cooke, 1988*). These excitatory and inhibitory synaptic connections could be differentially sensitive to pH. Additionally, although both networks are driven by bursting pacemaker neurons, the relative contribution of different ion channels to the burst generating mechanism may be sufficiently different so that the pacemakers themselves respond differently to high and low pH. A similar conclusion comes from a recent study that demonstrated that the arterioles in the retrotrapezoid nucleus, a region critical for respiratory chemosensation, and those in the cortex are differentially modulated during exposure to high $CO_2/H^+$ (*Hawkins et al., 2017*). Together, these findings support the possibility of specialization of a circuit's response to pH in the context of its local function.

The membrane potentials of the semi-isolated pyloric neurons, LP and PD, varied differentially with changes in extracellular pH. Semi-isolated LP neurons fired tonically and hyperpolarized in extreme base while semi-isolated PD neurons depolarized in acid. In intact preparations, we observed depolarization in acid for both neurons, suggesting an important role of circuit connectivity in regulating network activity across pH. These results illustrate the potential difficulty in generalizing the effects of environmental perturbation across neurons and circuits, even within the same animal.

Under most control conditions, the gastric mill rhythm was silent as is typically observed in STG preparations when the descending modulatory inputs are removed (*Hamood and Marder, 2015*). Unexpectedly, gastric mill rhythms were frequently activated upon exposure to or recovery from extreme pH. It is possible that either sensory or modulatory axons were activated by the pH changes, and it is feasible that specific sensory and/or modulatory neurons might be part of a circuit important in vivo for a response to altered pH.

In this study, we examined the effects of manipulating extracellular pH. However, the extent to which intracellular pH was affected and its contribution to changes in activity remain unclear. Neurons penetrated with intracellular recording electrodes exhibited more labile activity in response to changing pH than ones that were not penetrated with a sharp electrode. This may indicate that changes in intracellular pH would be more deleterious than what occurs in response to changes in extracellular pH alone. *Golowasch and Deitmer, 1993* revealed that extracellular pH in the STG of the crab, *Cancer pagurus*, was reliably around 0.1 pH more alkaline than bath pH while intracellular pH was 0.3 to 0.4 pH more acidic. Further, moderate shifts in bath pH – from pH 7.4 to 7.0 or 7.8 – resulted in negligible changes in pyloric frequency and slow and low amplitude shifts in extracellular pH while $NH_4Cl$ induced acidosis resulted in recoverable alkylosis of both the intracellular and extracellular space (*Golowasch and Deitmer, 1993*). These results suggest the restriction of the free diffusion of protons through the ganglion and the existence of active $Na^+$-dependent mechanisms to maintain more acidic intracellular and more alkaline extracellular compartments. *Golowasch and Deitmer, 1993* hypothesized that glial cells surrounding the neuronal processes in the neuropil of the STG may contain a $Na^+/H^+$ exchanger.

The results of *Golowasch and Deitmer (1993)* are consistent with what is known in other systems. The distribution of pH values within the brainstem during acute hypercapnia, an increase in blood $CO_2$, is heterogenous and location-dependent, reflecting local blood flow and pH regulatory

mechanisms (*Ichikawa et al., 1989*). Further, it remains uncertain as to whether or not pH sensation involved in the mammalian respiratory response is extracellular or intracellular (*Buckler et al., 1991*; *Fencl et al., 1966*; *Lassen, 1990*; *Ritucci et al., 1997*). In the pulmonate snail, *Helix aspersa*, the effect of hypercapnic solutions on the ventilatory response was greater than that of $NH_4Cl$ solutions of the same extracellular pH lacking $CO_2$, which suggests that the effects of $CO_2$ on intracellular pH is responsible for modulating chemoreception in this system (*Erlichman and Leiter, 1993*; *Erlichman and Leiter, 1994*). Mammalian studies have also suggested a role for glia in the modulation and maintenance of extracellular pH (*Kaila and Ransom, 1998*; *Ransom, 1992*; *Wenker et al., 2012*).

The ocean environment is both warming and acidifying at historic rates. *Cancer borealis* maintains relatively robust pyloric and cardiac rhythms in the temperature ranges it usually experiences (*Marder et al., 2015*; *Soofi et al., 2014*; *Tang et al., 2010*; *Tang et al., 2012*). The effect of temperature on ocean pH is relatively modest in comparison to the range of pH studied here; though, ocean pH can fluctuate in response to other environmental conditions. In *Carcinus maenas*, exposure to artificial ocean acidification produced relatively small changes in hemolymph pH (*Maus et al., 2018*). Therefore, unlike some ocean organisms that are very sensitive to even small ocean pH changes, we predict that the neuronal circuits in *C. borealis*, at least in adults, will be largely insensitive to changes in ocean pH, although the effects of pH swings may be exacerbated by changes in temperature or other environmental variables. Other physiological parameters, such as metabolic rates and hemolymph flow may be more pH sensitive (*Maus et al., 2018*). Moreover, network performance may be further attenuated when pH is coupled to increasing temperature and other environmental insults.

The data in this paper and in previous work on temperature reveal considerable animal-to-animal variability in response to extreme perturbations. Here, all preparations behaved predictably and reliably across more than a thousand-fold change in hydrogen ion concentration, an unexpectedly large range of robust performance. At more extreme pH, animal-to-animal variability became apparent, consistent with the responses of these circuits to extreme temperatures (*Marder et al., 2015*; *Soofi et al., 2014*; *Tang et al., 2010*; *Tang et al., 2012*). This animal-to-animal variability is almost certainly a consequence of the fact that similar network performance can arise from quite variable sets of underlying conductances (*Goaillard et al., 2009*; *Grashow et al., 2009*; *Grashow et al., 2010*; *Prinz et al., 2004*). What remains to be seen is whether animals that are more robust to a given extreme perturbation are less robust to others and whether there are given sets of network parameters that confer robustness to many different perturbations.

# Materials and methods

## Animals

From March 2016 to May 2018, adult male Jonah crabs (*Cancer borealis)* weighing between 400 and 700 grams were obtained from Commercial Lobster (Boston, MA). Before experimentation, all animals were housed in tanks with flowing artificial seawater (Instant Ocean) between 10°C and 13°C on a 12 hr light/dark cycle without food. Animals were kept in tanks for a maximum of two weeks. Animals were removed from tanks and kept on ice for 30 min prior to dissection.

## Saline solutions

Control *C. borealis* physiological saline was composed of 440 mM NaCl, 11 mM KCl, 13 mM $CaCl_2$, 26 mM $MgCl_2$, 11 mM Trizma base, and 5 mM Maleic acid. Additional quantities of concentrated HCl and NaOH were added to achieve solutions with pH 5.5, 6.0, 6.5, 7.0, 7.5, 8.0, 8.5, 9.0, 9.5, 10.0 at room temperature. Solutions were measured again at 11°C and determined to be at pH 5.5, 6.1, 6.7, 7.2, 7.8, 8.3, 8.8, 9.3, 9.8, and 10.4, respectively. Solution pH was measured using a calibrated pH/ion meter (Mettler Toledo S220). For experiments with picrotoxin, $10^{-5}$ M PTX was added to each of the pH solutions.

## Electrophysiology

The stomatogastric and cardiac nervous systems were dissected out of the animals and pinned out in a Sylgard (Dow Corning) coated plastic Petri dish containing chilled saline (11°C). In all cases, we

worked only with fully intact stomatogastric nervous system preparations that included the commissural and esophageal ganglia and their descending nerves. Only preparations containing healthy cardiac or pyloric rhythms with no sign of damage from dissection were analyzed.

Vaseline wells were placed around motor nerves and extracellular recordings were obtained using stainless steel pin electrodes placed in the wells and amplified using a differential amplifier (A-M Systems Model 1700). Intracellular sharp-electrode recordings were obtained from cell bodies in the stomatogastric ganglion using a microelectrode amplifier (Molecular Devices Axoclamp 2B or 900A) with HS-2A-x1LU headstages holding 15–30 MΩ borosilate microelectrodes with filaments (Sutter Instrument Co. BF150-86-10) pulled with a Flaming/Brown micropipette puller (Sutter Instrument Co. P-97). Microelectrodes were filled with a solution of 10 mM $MgCl_2$, 400 mM potassium gluconate, 10 mM HEPES, 15 mM $Na_2SO_4$, and 20 mM NaCl (*Hooper et al., 2015*).

Preparations were continuously superfused with physiological saline at 11°C. Superfusion was gravity fed at approximately 9 mL/min. The temperature of the superfusing saline was controlled and recorded using a Peltier device (Warner Instruments CL-100). Instantaneous bath pH was recorded using a pH microelectrode placed adjacent to the ganglion (Thermo Scientific Orion 9810BN) combined with a preamplifier (Omega PHTX-21). Output from the pH microelectrode was converted from arbitrary voltage to pH using a temperature-compensated calibration.

## Data acquisition and analysis

Data were acquired using a data acquisition board (Molecular Devices Digidata 1440A) and Clampex 10.5 software (Molecular Devices). Data were analyzed using Clampfit 10.5, Spike2 v 6.04 (Cambridge Electronic Design), and/or MATLAB 2017A (MathWorks). Figures were prepared in Adobe Illustrator CC 2017. All code is available for download at *Haley et al. (2018)* (copy archived at https://github.com/elifesciences-publications/haley_hampton_marder_2018).

For analyses of extracellular recordings of the *lvn* of the STG or the trunk of the CG, we analyzed the last 8 min of each 15 min pH step to ensure that extracellular pH had stabilized.

Data were categorized into states by manual annotation. A transition from one state to another was noted when there was a sustained change in activity lasting a minimum of 10 s. In other words, if the rhythm transitioned from one state into another and maintained the new state of activity for at least 10 s, a transition was noted at the start of that new state. The mean fraction of time that the preparations remained in each state during the last 8 min of recording at each pH step is plotted as stacked bar graphs.

Quantitative variables of frequency, number of spikes per burst, and duty cycle were measured using extracellular recordings. Spikes and bursts were first isolated in Spike2 by thresholding extracellular recordings. MATLAB was then used for further analysis. Instantaneous burst frequency was calculated by taking the inverse of the cycle's period, the time elapsed between the onset of one burst and the onset of the next. The number of spikes per burst of a given neuron reflects the number of spikes contributing to each burst. Duty cycle reflects the fraction derived by dividing the burst duration – time elapsed between the first and last spike – by the burst period. Mean values were computed for bins of 10 s such that for eight minutes of data, there were 48 binned mean values for each preparation, condition, and measure. Violin plots show distributions of these binned mean values pooled for all preparations. The body of the violin is a rotated kernel density estimate (KDE) plot. The circles give the median of the pooled data and the horizontal bars give the mean. The interquartile range (IQR) is given by the box plot within each violin with the whiskers giving the 95 percent confidence interval (CI).

For analyses of intracellular recordings of semi-isolated LP and PD neurons, the last minute of each pH step was analyzed in MATLAB. Minimum membrane potential was first measured by finding the minimum voltage of the neuron between each burst. Recordings were then low-pass filtered to remove spikes from the slow wave. Slow wave amplitude was measured by subtracting the trough from the peak of the slow wave's membrane potential. Spike amplitude was retrieved by subtracting the filtered slow wave signal from the original recording and then measuring the amplitude from trough to peak of each action potential. PD burst frequency was calculated by finding the inverse of the time period between one slow wave trough to the next. LP firing rate was determined by calculating the inverse of the inter-spike interval, the time between spikes. Mean values were computed for bins of 10 s. Violin plots show distributions of these binned mean values pooled for all preparations.

## Statistics

All statistics were performed using R (v 3.4.3). We performed statistical testing of the effects of acid and base on measures of the cardiac and pyloric rhythms using a Univariate Type III Repeated-Measures Analysis of Variance (ANOVA) from the car package. Separate tests were performed for acid and base step protocols. Post-hoc paired sample t-tests with Bonferroni correction were performed for each pH step against its respective control, the pH 7.8 condition immediately prior to the acid or base step protocol. To assess the differences between the effects of pH on the cardiac and pyloric rhythms, we performed a Two-Way Mixed-Measures ANOVA (Type III) for both acid and base step protocols using the car package. Post-hoc independent samples t-tests with Bonferroni correction were performed for each pH condition. T-tests were not performed for non-significant ANOVAs.

## Acknowledgements

We thank Daniel Shin and Katelyn Wadland for assistance in preparing and measuring pH solutions.

## Additional information

### Competing interests

Eve Marder: Deputy Editor, *eLife*. The other authors declare that no competing interests exist.

### Funding

| Funder | Grant reference number | Author |
|---|---|---|
| National Institutes of Health | R35 NS 097343 | Eve Marder |

The funders had no role in study design, data collection and interpretation, or the decision to submit the work for publication.

### Author contributions

Jessica A Haley, Data curation, Software, Formal analysis, Validation, Investigation, Visualization, Methodology, Writing—original draft, Writing—review and editing; David Hampton, Formal analysis, Investigation, Writing—original draft, Writing—review and editing; Eve Marder, Conceptualization, Resources, Data curation, Supervision, Funding acquisition, Project administration, Writing—review and editing

### Author ORCIDs

Jessica A Haley (iD) http://orcid.org/0000-0001-6282-7124
David Hampton (iD) http://orcid.org/0000-0001-8845-037X
Eve Marder (iD) http://orcid.org/0000-0001-9632-5448

### Decision letter and Author response

Decision letter https://doi.org/10.7554/eLife.41877.026
Author response https://doi.org/10.7554/eLife.41877.027

## Additional files

### Supplementary files

• Transparent reporting form
DOI: https://doi.org/10.7554/eLife.41877.022

### Data availability

Raw data files in .abf format are available for download at https://osf.io/r7aes/. All code is available for download https://github.com/jesshaley/haley_hampton_marder_2018 (copy archived at https://github.com/elifesciences-publications/haley_hampton_marder_2018). Source data files have been provided for Figures 2, 3, 4, 6, 7, 8, and their figure supplements.

The following dataset was generated:

| Author(s) | Year | Dataset title | Dataset URL | Database and Identifier |
|---|---|---|---|---|
| Haley JA, Hampton D | 2018 | Recordings of stomatogastric and cardiac ganglia of C. borealis at varying pH | https://doi.org/10.17605/osf.io/r7aes | Open Source Framework, 10.17605/OSF.IO/R7AES |

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
