## [Decision Letter]

Thank you for submitting your article "Two central pattern generators from the crab *C. borealis* respond robustly and differentially to extreme extracellular pH" for consideration by *eLife*. Your article has been reviewed by three peer reviewers, and the evaluation has been overseen by Ronald Calabrese as the Senior and Reviewing Editor. The following individuals involved in review of your submission have agreed to reveal their identity: Scott Hooper (Reviewer #1); Mark Beenhakker (Reviewer #2); Ronald M Harris-Warrick (Reviewer #3).

The reviewers have discussed the reviews with one another and the Reviewing Editor has drafted this decision to help you prepare a revised submission.

Summary:

Haley et al. present a compelling study that describes how pH differentially modulates two motor patterns generated by the crab nervous system. The study focuses on the pyloric rhythm and the cardiac rhythm, motor patterns generated by neurons in the stomatogastric ganglion and the cardiac ganglion, respectively. Two interesting phenomena are observed. First, both motor patterns are stable across an impressively broad range of pH's (6.1-8.8). Second, at extreme pH's (5.5 and 10.4), the pyloric and cardiac motor patterns are often altered, but in animal-specific ways. Generally, however, the pyloric rhythm was more sensitive to low pH while the cardiac rhythm was more sensitive to high pH.

The observation regarding motor pattern stability across pH is quite interesting as the ion channels expressed by the constituent motor circuit neurons are likely pH-sensitive. While the authors do not provide direct evidence of such sensitivity, many previous studies, including those of the stomatogastric ganglion, show that ion channel conductances can be pH-dependent. Considering the strong likelihood that channel activity is changing, it is surprising that the motor output is unaltered.

In sum, the authors present an interesting study that carefully documents the pH sensitivity of motor circuits in the crab. The inclusion of corresponding data that demonstrates the pH sensitivity of the underlying ionic conductances would be nice, but is likely beyond the scope of the manuscript. The opinion that the study is thorough and includes an assessment of two independent motor circuits in the same animal tempers this critique.

Essential revisions:

There were several concerns that need to be addressed in a comprehensive revision of the manuscript.

1) Warming of the oceans is used as a basis for arguing that crabs are likely to experience pH stress. This argument about temperature cannot account for the broad range of pH tolerated. Indeed it seemed to the reviewers that much of the pH range explored is artificial. It is better to frame the study as extending well beyond any likely physiological pH range (~6.8 – 8.8) and focus Introduction and Discussion on how such pH tolerance might be achieved. Possibilities might include:

a) Mammalian proteins and similar have become very pH sensitive in evolution, and the primitive state of these entities was one of much less pH sensitivity.

b) pH sensitivity is indeed the general state of proteins and similar, and crabs have been for some reason evolved so that their neuronal networks were pH resistant. This could occur due to the individual molecules being pH resistant, or arise from "balancing" compensatory interactions in the ensemble of entities underlying neuron physiology. Then lead into how these data should inform future work on these issues. A major rewrite of Introduction and Discussion is in order.

2) There was a concern that the pH was adjusted in a non-physiological way (i.e., using strong acids and bases not in a CO2 HCO_3_^-^ system) and thus providing challenges to pumps.

3) Several other concerns are to be found in the detailed reviews that can be answered in the context of these two major points.

*Reviewer #1 Minor Comments:*

Some of the networks (Figure 2) recorded from in the present work continue to produce relatively normal rhythms over a pH range from 5.5 to 10.4. To put this range into perspective, [H+] at pH 5.5 is 3.2 x 10-6 M and at pH 10.4 4 x 10-11 M, and thus an 80,000-fold change in [H+] [Note error that mentions a "three-fold range of hydrogen ion concentration" whereas the actual change in [H+] for the figure in question over which rhythmic pyloric activity continued is a change from pH 6.7 to 8.8, which correspond to [H+]'s of 2 x 10-7 and 1.6 x 10-9, a 125-fold change].

Text relating to the PTX-applied experiments: these are not isolated neurons, since the cholinergic neurons were not photo-inactivated. Please find another word or phrase to describe these neurons.

*Reviewer #2:*

1) Discussion, third paragraph: A study by Meuth et al., 2006, seems relevant. Therein, the authors show that the membrane potential of thalamocortical neurons remains unaltered across a range of pH's, a stability attributed to the counterbalancing activities of TASK and HCN channels [Meuth et al., 2006].

2) To what extent is it known that pyloric and cardiac neurons express similar channels? Of those, are there known pH-sensitive conductances?

3) Are the differential responses of the pyloric and cardiac rhythms consistent with feeding behaviors?

4) In the Abstract, the authors suggest that "animal-to-animal variability was likely a consequence of similar network performance arising from variable sets of underlying conductances". I'm having difficulty understanding this statement, perhaps because it's not clear what "variability" is referring to. Is the point that despite similar network performance across the population, the motor pattern responses to pH from one animal to the next are quite variable because the neurons of each crab likely express different sets of ionic conductances, therefore endowing each cell with a unique pH sensitivity that may or may not affect circuit activity?

5) I'm struggling a bit to understand the take home message the authors are trying to convey. The Abstract might suggest a cautionary message regarding the difficulty of generalizing the effects of environmental perturbations. The beginning of the Introduction might suggest a cautionary message regarding changing ocean temperatures/pH, but that the crabs will be okay (end of Discussion). The end of the Introduction might suggest a message regarding impressive network stability across a broad range of pHs.

6) Regarding motor pattern alteration at extreme pH's, it is intriguing that one does not simply observe a general breakdown of motor circuit function. Do the collective motor pattern responses make behavioral sense? That is, if the cardiac rhythm changes, should the pyloric rhythm remain stable, and vice versa?

Reviewer #3 Minor Comments:

This reviewer was concerned that most of the description in the Results focuses on responses to extremely non-physiological pH changes, and thus chase experimental artifacts with less attention to the probable natural pH range to be encountered.

1) The range of pH tested is truly tremendous, reflecting nearly 10,000-fold variation in the concentration of H^+^. As the Introduction rationalized this study on the possible effect of a warming ocean on pH, it would be rational to vary the pH within some range relating to the real experience of crabs (similar to the temperature ranges in other work from the Marder lab). If we allow an extreme 40 °C range variation, this would correspond to H^+^ variation by 0.64 pH units. Clearly, from the data presented, there are no effects of pH within any reasonable natural range, say 7.5 to 8.1.

2) While the Abstract and Discussion both emphasize this point, and reflect on possible mechanisms for stabilization of the motor patterns over a large pH range, the results are not focused on this important point; instead they focus primarily on changes at the pH extremes (below 6.1 or above 9.3), which are far outside any range the system was evolved to deal with. These are certainly experimental artifacts, similar, for example, to changing the circuit function when the concentration of Ca^2+^ is altered over the range from 0.1 mM to 1500 mM. Better to focus on the main result, which is the stability of the circuit patterns over any possible natural pH range.

3) The results are descriptive, with no attempt to address the mechanisms by which stability is maintained as pH is changed. Many possible explanations are discussed in the Discussion, but none are tested.

---

## [Author Response]

There were several concerns that need to be addressed in a comprehensive revision of the manuscript.1) Warming of the oceans is used as a basis for arguing that crabs are likely to experience pH stress. This argument about temperature cannot account for the broad range of pH tolerated. Indeed it seemed to the reviewers that much of the pH range explored is artificial. It is better to frame the study as extending well beyond any likely physiological pH range (~6.8 – 8.8) and focus Introduction and Discussion on how such pH tolerance might be achieved. Possibilities might include:a) Mammalian proteins and similar have become very pH sensitive in evolution, and the primitive state of these entities was one of much less pH sensitivity.b) pH sensitivity is indeed the general state of proteins and similar, and crabs have been for some reason evolved so that their neuronal networks were pH resistant. This could occur due to the individual molecules being pH resistant, or arise from "balancing" compensatory interactions in the ensemble of entities underlying neuron physiology. Then lead into how these data should inform future work on these issues. A major rewrite of Introduction and Discussion is in order.

We have reframed the paper. Our original impetus for doing this work was not so much to understand the possible effects of ocean acidification but to probe the consequences of mild and extreme perturbations on the circuits of the stomatogastric and cardiac ganglia. Consequently, we have removed the paragraph on ocean acidification from the Introduction and have reframed parts of the Discussion to better frame our study within this context.

2) There was a concern that the pH was adjusted in a non-physiological way (i.e., using strong acids and bases not in a CO2 HCO_3_^-^ system) and thus providing challenges to pumps.

We understand the reviewers’ concern. We thought hard about the best way to manipulate the pH of our system. As there is no way to make a bicarbonate buffer work over the pH range that we wanted to test, we decided to use a strong acid and base to push the saline into these ranges. We made this decision in an effort to maintain the saline composition (440 mM NaCl, 11 mM KCl, 13 mM CaCl_2_, 26 mM MgCl2, 11 mM Trizma base, and 5 mM Maleic acid) that has been well studied. Given that we were using pH as a perturbation to explore the robustness and variability of these circuits to perturbation, we don’t believe that this method of pH adjustment should affect the interpretation of our results.

Further, this study revealed surprisingly robust activity of these circuits in the pH altered salines. If these preparations had been more sensitive than expected to pH, we would have been concerned that the effects could be partially attributable to how the pH was manipulated. Given that these circuits are highly robust and that we were able to explore a large range of pH without significantly altering the control saline, we believe that the potential effects of strong acids and bases on pumps do not impact the results and conclusions of this study.

Reviewer #1 Minor Comments:

Some of the networks (Figure 2) recorded from in the present work continue to produce relatively normal rhythms over a pH range from 5.5 to 10.4. To put this range into perspective, [H+] at pH 5.5 is 3.2 x 10-6 M and at pH 10.4 4 x 10-11 M, and thus an 80,000-fold change in [H+] [Note error that mentions a "three-fold range of hydrogen ion concentration" whereas the actual change in [H+] for the figure in question over which rhythmic pyloric activity continued is a change from pH 6.7 to 8.8, which correspond to [H+]'s of 2 x 10-7 and 1.6 x 10-9, a 125-fold change].

We thank the reviewer for the correction and have revised the error.

Text relating to the PTX-applied experiments: these are not isolated neurons, since the cholinergic neurons were not photo-inactivated. Please find another word or phrase to describe these neurons.

We have changed the wording to “semi-isolated”, clarifying that under PTX, this circuit lacks only glutamatergic synapses.

Reviewer #2:

1) Discussion, third paragraph: A study by Meuth et al., 2006, seems relevant. Therein, the authors show that the membrane potential of thalamocortical neurons remains unaltered across a range of pH's, a stability attributed to the counterbalancing activities of TASK and HCN channels [Meuth et al., 2006].

We thank the reviewer. It is indeed relevant and have added a reference to these experiments. See Discussion, fourth paragraph.

2) To what extent is it known that pyloric and cardiac neurons express similar channels? Of those, are there known pH-sensitive conductances?

There is almost a complete overlap of ion channel genes expressed in the two ganglia, to the extent that it is difficult to determine cell identity solely through expression profiles of their channels and receptors (Northcutt et al., 2018). We have revised the text to make this comparison clearer. See Discussion, fifth paragraph.

3) Are the differential responses of the pyloric and cardiac rhythms consistent with feeding behaviors?

As the reviewers pointed out, over the physiologically relevant pH range both ganglia were relatively insensitive to pH, therefore we think that the relative sensitivity of the two ganglia may not be behaviorally relevant, but may speak more to underlying differences in synaptic connectivity and circuit organization.

4) In the Abstract, the authors suggest that "animal-to-animal variability was likely a consequence of similar network performance arising from variable sets of underlying conductances". I'm having difficulty understanding this statement, perhaps because it's not clear what "variability" is referring to. Is the point that despite similar network performance across the population, the motor pattern responses to pH from one animal to the next are quite variable because the neurons of each crab likely express different sets of ionic conductances, therefore endowing each cell with a unique pH sensitivity that may or may not affect circuit activity?

The reviewer was correct; this sentence was unclear. We have revised it accordingly.

5) I'm struggling a bit to understand the take home message the authors are trying to convey. The Abstract might suggest a cautionary message regarding the difficulty of generalizing the effects of environmental perturbations. The beginning of the Introduction might suggest a cautionary message regarding changing ocean temperatures/pH, but that the crabs will be okay (end of Discussion). The end of the Introduction might suggest a message regarding impressive network stability across a broad range of pHs.

We have reworded the Abstract, Introduction, and Discussion to clarify the message.

6) Regarding motor pattern alteration at extreme pH's, it is intriguing that one does not simply observe a general breakdown of motor circuit function. Do the collective motor pattern responses make behavioral sense? That is, if the cardiac rhythm changes, should the pyloric rhythm remain stable, and vice versa?

We were actually surprised to see the differential sensitivity of the two ganglia to acid and base. In another study we have recorded non-invasively from the heart and pyloric muscles and seen that the two organs are only weakly correlated in activity. Moreover, the heart sometimes stops entirely for minutes while the pyloric rhythm continues. Therefore, we were not surprised that there wasn’t a general breakdown of motor circuit function.

Reviewer #3 Minor Comments:

This reviewer was concerned that most of the description in the Results focuses on responses to extremely non-physiological pH changes, and thus chase experimental artifacts with less attention to the probable natural pH range to be encountered.1) The range of pH tested is truly tremendous, reflecting nearly 10,000-fold variation in the concentration of H^+^. As the Introduction rationalized this study on the possible effect of a warming ocean on pH, it would be rational to vary the pH within some range relating to the real experience of crabs (similar to the temperature ranges in other work from the Marder lab). If we allow an extreme 40 °C range variation, this would correspond to H^+^ variation by 0.64 pH units. Clearly, from the data presented, there are no effects of pH within any reasonable natural range, say 7.5 to 8.1.

We have reworded the Introduction to better reflect our motivations for this study.

2) While the Abstract and Discussion both emphasize this point, and reflect on possible mechanisms for stabilization of the motor patterns over a large pH range, the results are not focused on this important point; instead they focus primarily on changes at the pH extremes (below 6.1 or above 9.3), which are far outside any range the system was evolved to deal with. These are certainly experimental artifacts, similar, for example, to changing the circuit function when the concentration of Ca^2+^ is altered over the range from 0.1 mM to 1500 mM. Better to focus on the main result, which is the stability of the circuit patterns over any possible natural pH range.

We have reframed the paper and believe that the results faithfully report our observations in the context of this study.

3) The results are descriptive, with no attempt to address the mechanisms by which stability is maintained as pH is changed. Many possible explanations are discussed in the Discussion, but none are tested.

Understanding the mechanisms would be very interesting, but is beyond the scope of this first paper.